# Effectiveness of BMP-2 and PDGF-BB Adsorption onto a Collagen/Collagen-Magnesium-Hydroxyapatite Scaffold in Weight-Bearing and Non-Weight-Bearing Osteochondral Defect Bone Repair: In Vitro, Ex Vivo and In Vivo Evaluation

**DOI:** 10.3390/jfb14020111

**Published:** 2023-02-16

**Authors:** Jietao Xu, Shorouk Fahmy-Garcia, Marinus A. Wesdorp, Nicole Kops, Lucia Forte, Claudio De Luca, Massimiliano Maraglino Misciagna, Laura Dolcini, Giuseppe Filardo, Margot Labberté, Karin Vancíková, Joeri Kok, Bert van Rietbergen, Joachim Nickel, Eric Farrell, Pieter A. J. Brama, Gerjo J. V. M. van Osch

**Affiliations:** 1Department of Orthopedics and Sports Medicine, Erasmus MC, University Medical Center, 3015 GD Rotterdam, The Netherlands; 2Department of Oral and Maxillofacial Surgery, Erasmus MC, University Medical Center, 3015 GD Rotterdam, The Netherlands; 3Fin-Ceramica Faenza S.p.A, 48018 Faenza, Italy; 4Applied and Translational Research Center, IRCCS Rizzoli Orthopaedic Institute, 40136 Bologna, Italy; 5School of Veterinary Medicine, University College Dublin, D04 V1W8 Dublin, Ireland; 6Department of Biomedical Engineering, Eindhoven University of Technology, 5612 AZ Eindhoven, The Netherlands; 7Department Tissue Engineering and Regenerative Medicine, University Hospital Würzburg, 97070 Würzburg, Germany; 8Department of Otorhinolaryngology, Erasmus MC, University Medical Center, 3015 GD Rotterdam, The Netherlands; 9Department of Biomechanical Engineering, Delft University of Technology, 2628 CD Delft, The Netherlands

**Keywords:** tissue engineering, regenerative medicine, osteochondral lesion, biocompatible materials, bone morphogenetic proteins, platelet-derived growth factor, animal model, weight-bearing

## Abstract

Despite promising clinical results in osteochondral defect repair, a recently developed bi-layered collagen/collagen-magnesium-hydroxyapatite scaffold has demonstrated less optimal subchondral bone repair. This study aimed to improve the bone repair potential of this scaffold by adsorbing bone morphogenetic protein 2 (BMP-2) and/or platelet-derived growth factor-BB (PDGF-BB) onto said scaffold. The in vitro release kinetics of BMP-2/PDGF-BB demonstrated that PDGF-BB was burst released from the collagen-only layer, whereas BMP-2 was largely retained in both layers. Cell ingrowth was enhanced by BMP-2/PDFG-BB in a bovine osteochondral defect ex vivo model. In an in vivo semi-orthotopic athymic mouse model, adding BMP-2 or PDGF-BB increased tissue repair after four weeks. After eight weeks, most defects were filled with bone tissue. To further investigate the promising effect of BMP-2, a caprine bilateral stifle osteochondral defect model was used where defects were created in weight-bearing femoral condyle and non-weight-bearing trochlear groove locations. After six months, the adsorption of BMP-2 resulted in significantly less bone repair compared with scaffold-only in the femoral condyle defects and a trend to more bone repair in the trochlear groove. Overall, the adsorption of BMP-2 onto a Col/Col-Mg-HAp scaffold reduced bone formation in weight-bearing osteochondral defects, but not in non-weight-bearing osteochondral defects.

## 1. Introduction

Osteochondral tissue is formed by two main tissue types: the articular cartilage, which functions as a low-friction and wear-resistant surface, and the subchondral bone, which plays a crucial mechanically supportive role [1]. Without effective and timely interventions, damage to this osteochondral unit, caused by traumatic injury or disease, may progress to osteoarthritis [2,3]. Bi-layered biomaterial scaffolds have been developed to restore the structural and physiological properties of the entire osteochondral unit and, thus, to support chondrogenesis and osteogenesis simultaneously [4,5]. As a biologically derived protein, collagen is an efficient biomaterial to support cellular activities and promote osteochondral repair [6,7,8]. The addition of hydroxyapatite (HAp) can improve the osteogenic potential of a collagen-based scaffold in vivo, and magnesium ions (Mg^2+^) induce osteogenic differentiation and osteoblast differentiation [9]. Preclinically, bi-layered collagen/collagen-magnesium-HAp (Col/Col-Mg-HAp) scaffolds have successfully reconstructed the articular cartilage and the subchondral bone in animal models [10,11,12]. Clinical cohort studies also demonstrated the excellent stability of this scaffold and clinical improvement in knee function [13,14,15]. However, subchondral bone repair remained suboptimal in some clinical follow-ups in comparison to the cartilage repair capacity of this scaffold [15], which may lead to altered biomechanical properties of the osteochondral unit and thereby affect the long-term survival of the neo-cartilage [16]. This might subsequently lead to renewed osteochondral damage and joint disease [17,18].

Incorporating factors that stimulate bone formation could be a promising approach to overcome this limitation in bone regeneration capacity [19]. Bone morphogenetic protein 2 (BMP-2) has a vital role in osteogenesis and osteoclastogenesis [20,21] and is approved by the Food and Drug Administration (FDA) as an osteogenic protein. The recruitment of stem cells or osteoblasts is necessary for osteogenic initiation, and BMP-2 was reported to facilitate cell ingrowth [22]. Platelet-derived growth factor (PDGF) is potent in stimulating cell ingrowth, angiogenesis, and osteogenesis [23,24,25,26]. The delivery of BMP-2 and PDGF onto biomaterials has been shown to provide an improvement in osteoblast function and bone integration [27,28,29]. Therefore, it might be promising to adsorb BMP-2 and PDGF-BB onto this Col/Col-Mg-HAp scaffold to improve bone healing.

This study aimed to evaluate the osteogenic effectiveness of BMP-2 or PDGF-BB adsorption onto a Col/Col-Mg-HAp scaffold on bone repair in osteochondral defects. We first assessed the release profiles of BMP-2 and PDGF-BB from the two layers of a Col/Col-Mg-HAp scaffold in vitro. Next, an ex vivo osteochondral culture model was used to investigate the added effect of growth factors on cell ingrowth from adjacent tissues. Then, we investigated the effect of BMP-2 or PDGF-BB incorporated into a Col/Col-Mg-HAp scaffold in an in vivo semi-orthotopic mouse model for the early phases of tissue repair. Finally, the effect of BMP-2 incorporated into a Col/Col-Mg-HAp scaffold was investigated in both weight-bearing and non-weight-bearing locations of the knee joint in an established preclinical caprine osteochondral defect model.

## 2. Materials and Methods

### 2.1. Scaffold Fabrication and Characterization

Col/Col-Mg-HAp scaffold is a biomimetic scaffold that has a porous, 3-dimensional composite structure. The scaffold is composed of two layers: the cartilaginous layer, consisting of Type I collagen, to maintain joint congruence and the bone layer consisting of a combination of Type I collagen (60%) and magnesium-hydroxyapatite (40%). Each layer of the scaffold is synthesised separately by a standardised process from an atelocollagen aqueous solution (1% *w/w*) in acetic acid, isolated from equine tendon. The upper non-mineralised chondral layer of the scaffold is obtained by dissolving an acetic solution of Type I collagen in bi-distilled water by adding NaOH. The bone layer of the scaffold is obtained by nucleating nanostructured hydroxyapatite into self-assembling collagen fibres, as occurs in the natural biological neo-ossification process. To stabilise the scaffold, the fibrous structures were chemically cross-linked for 16 hours at room temperature. After chemical cross-linking, the two layers were superimposed and afterwards they are freeze-dried.

The morphology of the scaffold was evaluated by Scanning Electron Microscopy (SEM) performed on a SEM-LEO 438 VP (Carl Zeiss AG, Oberkochen, Germany). The samples were sputter coated with gold prior to examination. The mineral content of the bone layer was evaluated by thermogravimetric analysis (TGA), performed in alumina crucibles in an air atmosphere with a flow rate of 80 mL/min, between 25 and 700 °C (Mettler Toledo DT-TGA/DSC1 Star System, Columbus, OH, USA). The elemental composition of the mineral phase (magnesium-hydroxyapatite) was determined using inductively coupled plasma-optical emission spectrometry (ICP-OES, Thermo Scientific iCAP 7400, Waltham, MA, USA). In particular, the composition is expressed as Ca/P, (Ca+Mg)/P, Mg/Ca% molar ratios. The bone layer was dissolved in hot nitric acid (65 *v/v*%) in order to completely destroy the collagen matrix and solubilise the inorganic phase. The content (ppm) of magnesium, calcium, and phosphorous in the samples is determined by comparison with a predetermined standard curve: Ca/P = 1.5 ± 0.1%; (Ca+Mg)/P = 1.6 ± 0.1%; Mg/Ca% = 1.5 ± 0.4%. The total porosity of the osteochondral scaffold was determined using Archimedes’ principle. The exterior volume (V_s_) of the sample was measured using a Vernier calliper. The sample was then immersed in a pycnometer containing 96% ethanol solution. The actual volume (V_a_) of the sample is calculated using the formula:$$V_{a}=\frac{\left(W_{W}-W_{o}\right)-\left(W_{t}-W_{p}\right)}{0.789}\;g/{\mathrm{cm}}^{3}$$

W_w_ is the weight of the ethanol and the pycnometer; W_o_ is the dry weight of the pycnometer; W_t_ is the combined weight of the ethanol, the pycnometer, and the plug sample; W_p_ is the combined weight of the dry pycnometer and dry plug sample; and 0.789 g/cm^3^ is the density of ethanol solution. The porosity of the scaffold was then determined using the following formula:$$\mathrm{Total}\;\mathrm{Porosity}\left(\%\right)=\frac{V_{s}-V_{a}}{V_{s}}*100$$

### 2.2. BMP-2 and PDGF-BB Release from the Different Layers of a Col/Col-Mg-HAp Scaffold

To investigate the release kinetics of BMP-2 and PDGF-BB from the different layers of the Col/Col-Mg-HAp scaffold, a time course study was performed (Appendix A). A quantity of 35 μL (28.5 μg/mL) BMP-2 or PDGF-BB (Sigma, Saint Louis, MI, USA) was absorbed into the separated layers (either collagen-only layer or Col-Mg-HAp layer) of a Col/Col-Mg-HAp scaffold (Osteochondral scaffold, Finceramica, Italy) in a low-affinity binding plate at 37 degrees Celsius for 30 min. The concentration of growth factors was determined according to our previous studies [30,31]. After absorption, the medium was harvested from the plate, and the scaffolds were transferred to a new low-affinity binding plate. A quantity of 800 μL alpha-Minimum Essential Medium (α-MEM, Gibco, Waltham, MA, USA) was added to each scaffold-containing well. At each time point (6, 24, 48, 72, 96, 120, 144, 168 h, 336 h only for BMP-2), the medium was collected and replaced by fresh medium. The collected medium was analysed for BMP-2 by recombinant human BMP-2 (Peprotech, Cranbury, NJ, USA) or PDGF-BB by a recombinant human PDGF-BB DuoSet ELISA kit (R&D Systems, McKinley Place N.E., Minneapolis, MN, USA) according to the manufacturer’s instructions.

### 2.3. Cell Recruitment Capacity of BMP-2 or PDGF-BB in an Ex Vivo Osteochondral Defect Culture Model

To study the effect of BMP-2 and PDGF-BB adsorbed onto a bi-layered Col/Col-Mg-HAp scaffold on cell recruitment capacity, an ex vivo osteochondral defect culture model, previously developed and validated in our laboratory, was used [32] (Appendix A). Briefly, osteochondral defects were created in bovine osteochondral biopsies (8 mm diameter, 5 mm height) harvested from metacarpal-phalangeal joints of 6- to 8-month-old calves (LifeTec, Eindhoven, The Netherlands), in which a 4 mm wide and 4 mm high defect was created. The osteochondral plugs were kept overnight in Dulbecco’s Modified Eagle Medium high glucose (DMEM, 4.5 g/L glucose, Gibco, Waltham, MA, USA) supplemented with 10% fetal bovine serum (FBS, Gibco, Waltham, MA, USA), 50 μg/mL gentamycin (Gibco, Waltham, MA, USA), and 1.5 μg/mL fungizone (Gibco, Waltham, MA, USA). The following day, the Col/Col-Mg-HAp scaffolds (diameter: 4 mm, height: 4 mm) with or without adsorbed growth factors (*n* = 4 for each condition) were inserted into the osteochondral defects. Either 4 μg (57.1 μg/mL) BMP-2 solution or 100 ng (1.4 μg/mL) PDGF-BB was adsorbed onto the scaffold. Each construct was cultured in 3.5 mL medium in a new 12-well plate at 37 °C and 5% CO_2_. The medium was refreshed on the first day and subsequently every two days. After 3 weeks, the osteochondral constructs were harvested and fixed in 4% formalin for 1 week with subsequent further histological analysis.

### 2.4. In Vivo Osteochondral Defect Model in Mice

To assess the effect of BMP-2 and PDGF-BB in the Col/Col-Mg-HAp scaffolds on bone repair, an established in vivo subcutaneous mouse model developed previously in our laboratory was used (Appendix A) [33]. Osteochondral biopsies were harvested, and the defects were created as described previously (see under Section 2.3) and kept overnight in α-MEM supplemented with 10% FBS, 50 μg/mL gentamycin, and 1.5 μg/mL fungizone until implantation. Eleven 12-week-old NMRI-Fox1nu mice (Taconic, Albany, NY, USA) were used for this study. The animals were randomly assigned and housed under specific-pathogen-free conditions with a regular day/night light cycle and allowed to adapt to the conditions of the animal facility for 7 days. Food and water were available ad libitum. Before implantation, 70 µL of saline solution, or 70 µL of saline solution containing BMP-2 (57.1 μg/mL, 4 μg) or PDGF-BB (28.5 μg/mL, 2 μg or 1.4 μg/mL, 100 ng) was added dropwise onto the Col/Col-Mg-HAp scaffolds. All osteochondral plugs were covered with a circular 8 mm Neuro-Patch membrane (Braun, Melsungen, Germany) to prevent the ingrowth of host cells from the top. The osteochondral plugs were randomly implanted in subcutaneous pockets on the back of the mice under 2.5–3% isoflurane anaesthesia (1000 mg/g, Laboratorios Karizoo, Maharashtra, India). One osteochondral plug was implanted per pocket, and four osteochondral plugs were implanted per mouse. The incisions were closed with staples (Fine Science Tools, Vancouver, BC, Canada). At 1 h before surgery and at 6–8 h after surgery, 0.05 mg/kg body weight of buprenorphine (Chr. Olesen & Co, Gentofte, Copenhagen, Denmark) was injected subcutaneously to ensure pre- and postoperative analgesia. Mice received a subcutaneous prophylactic antibiotic injection of 25 mg/kg body weight of Amoxicillin (Dopharma, Raamsdonksveer, The Netherlands).

After 4 or 8 weeks, mice were euthanised by cervical dislocation under 2.5–3% isoflurane anaesthesia, and the osteochondral plugs were harvested. All samples were fixed in 4% formalin for 1 week for further analysis. This animal experiment was approved by the Ethics Committee for Laboratory Animal Use (AVD101002016991; protocol #EMC 16-691-05).

### 2.5. In Vivo Osteochondral Defect Caprine Model

A validated bilateral osteochondral defect caprine model was used to assess the osteochondral defect repair capacity of BMP-2-supplemented scaffolds in a preclinical large animal model (Appendix A). An experimental unit of 11 skeletally mature female Saanen goats (age: 3 years, weight: 37.9 ± 7.3 kg) was subjected to a bilateral arthrotomy under general anaesthesia as described before [34,35,36]. In short: all animals received a prophylactic antibiotic injection with amoxycillin clavulanic acid 8.75 mg/kg intramuscular (Noroclav, Norbrook, Ireland) and were intravenously sedated with butorphanol (0.2 mg/kg, Butador, Chanelle Pharma, Ireland) and diazepam (0.2 mg/kg, Diazemuls; Accord Healthcare, UK). A lumbosacral epidural block with lidocaine (2 mg/kg, Lidocaine HCl 2%, B. Braun Medical Inc., EU, Melsungen, Germany) and morphine (0.2 mg/kg, Morphine Sulphate 10 mg/mL, Kalceks, Latvia) was performed with the animal in sternal recumbency. Anaesthesia was induced with propofol IV to effect (max. 6 mg/kg, Propofol-Lipuro 1%, B. Braun Medical Inc., Melsungen, German) and was maintained with isoflurane (Vetflurane, Virbac Animal Health, Suffolk, UK) in 100% oxygen via a circle rebreathing system. All animals received analgesia with meloxicam IV (0.5 mg/kg, Rheumocam, Chanelle, Galway, Ireland); and morphine IV (0.2 mg/kg, Morphine sulphate, Mercury Pharmaceuticals, Dublin, Ireland) 90 min after the epidural block.

An arthrotomy of each stifle joint was performed in dorsal recumbency using a lateral parapatellar approach. Under constant irrigation with saline, a pointed 6 mm drill bit was used to drill an approximate 3–4 mm deep non-weight-bearing defect in the transition of the distal 1/3 to the middle 1/3 of the trochlear groove and in the weight-bearing part of the medial femoral condyle. Subsequently, a custom-made flattened drill bit and a depth guide were used to create an exact flat 6 mm deep by 6 mm wide circular critical-sized osteochondral defect in a non-weight-bearing and a weight-bearing location. The joint was flushed with saline to remove any debris, and the defects were press fit with a similar-sized selected scaffold before surgical closure as described before. Each stifle joint was randomly assigned to one of the two treatment groups (Appendix A): (1) Col/Col-Mg-HAp scaffold-only (6 mm diameter, 6 mm height, Osteochondral scaffold, Finceramica, Italy), and (2) Col/Col-Mg-HAp scaffold adsorbed with BMP-2 (57.1 μg/mL).

Following surgery, postoperative analgesia was provided (meloxicam 5 days) and goats were housed in indoor pens for daily postoperative welfare monitoring and scoring. Two weeks postoperatively, following the removal of skin sutures, animals were released to pasture or loose housing (weather dependent) for the remainder of the study period with daily health checks. An orthopaedic assessment (Appendix A) was performed on the day of humane euthanasia under sedation with a barbiturate overdose at the predetermined endpoint at 6 months after surgery. Subsequently, all the joints, surrounding joint tissues, and synovial fluids were scored (Appendix A), dissected, and photographed (Body Canon EOS R5, lens: Canon EF 100 mm f/2.8 L Macro IS USM, flash: Macro Ring lite MR-14EX II). Biopsies 1 cm by 1 cm square containing the entire defects were harvested with an oscillating saw.

Ethical evaluation and approval were provided by the Health Products Regulatory Authority of Ireland (AE1898217/P032), the Animal Research Ethics Committee of University College Dublin (AREC-P-12–71) and the Lyons Animal Welfare Board (Health, Husbandry and Monitoring plans).

### 2.6. Macroscopic Assessment of the Defect Repair in the Caprine Model

The quality of the cartilage repair in the caprine samples was assessed semi-quantitatively using the International Cartilage Repair Society (ICRS) macroscopic evaluation system (Appendix A) [37] and a macroscopic scoring system (Appendix A) developed by Goebel et al. [38]. The ICRS scoring system evaluates the macroscopic appearance of cartilage repair tissue as Grade IV (severely abnormal), Grade III (abnormal), Grade II (nearly normal), or Grade I (normal). The Goebel Score describes macroscopic articular cartilage repair with five major evaluation categories. The quality of defect repair was scored blinded on fresh samples by two independent assessors, and the scores were averaged for further analysis. All the samples were fixed in 4% formalin for 10 days after macroscopic assessment for further analysis.

### 2.7. Micro-Computed Tomography

From the mouse model, the retrieved bovine osteochondral plugs were scanned (Quantum GX, Perkin Elmer, Akron, OH, USA) with the following settings after fixation in 4% formalin: energy 90 KV, intensity 88 μA, 18 mm FOV, 36 μm isotropic voxel size. All the scans above were under an X-ray filter of Cu (thickness = 0.06 mm) and Al (thickness = 0.5 mm) and were calibrated using a phantom with a known density of 0.75 g/cm^3^, which was additionally scanned before and after each scan. A high-resolution mode was set, and a scan time of 4 min was used.

The caprine samples were scanned with the same settings except for 36 mm FOV, 72 μm isotropic voxel size. Image processing included modest Gauss filtering (sigma = 0.8 voxel, width = 1 voxel) and segmentation using a single threshold. A cylindrical region (4 mm diameter and 5 mm height) within the original defect (6 mm diameter and 6 mm height) was selected as a volume of interest (VOI) for the caprine samples. In this VOI, the following morphometric parameters were measured: bone volume per total volume (BV/TV), trabecular thickness (Tb.Th), trabecular number (Tb.N), and trabecular separation (TB.Sp). Morphological analyses were performed using IPL (Scanco Medical AG, Bruettisellen, Switzerland).

### 2.8. Histology and Immunohistochemistry

The bovine osteochondral plugs cultured ex vivo were decalcified for 2 weeks using 10% formic acid (Sigma, Saint Louis, MI, USA). After micro-CT scanning, the bovine osteochondral plugs harvested from mice were decalcified using 10% ethylenediaminetetraacetic acid (EDTA, Sigma, Saint Louis, MI, USA) for 4 weeks. The caprine samples were decalcified for 3 weeks using 10% formic acid. Subsequently, all samples were embedded in paraffin and sectioned at 6 µm. Following dewaxing, H&E staining was performed with Hematoxylin (Sigma, Saint Louis, MI, USA) and Eosin Y (Merck, Kenilworth, NJ, USA) to study general cell and tissue morphology. To visualise glycosaminoglycans in the extracellular matrix (ECM), dewaxed sections were stained with Safranin O (Fluka, Buchs, Switzerland) and Light green (Fluka, Buchs, Switzerland). To study the regenerated tissue type in the osteochondral defects, RGB staining was performed (Proteoglycans/hyaline cartilage appears blue, mineralised cartilage matrix appears pink/greenish, collagen fibres/uncalcified bone appears red, and mineralised bone appears green) using Alcian Blue (Sigma, Saint Louis, MI, USA), Fast Green (Sigma, Saint Louis, MI, USA), and Picrosirius Red (Sigma, Saint Louis, MI, USA) [39]. The cell number in the scaffolds was counted under microscopy. NDP View2 software (version 2.8.24, 2020 Hamamatsu Photonics K.K.) was used to measure the tissue volume in the defect at three sections that were taken at the middle, 0.5 mm, and 1 mm further away for bovine samples or at the middle for caprine samples (Appendix A). The percentage of the defect covered with newly formed osteochondral tissue (100% indicated that the defect was fully filled with newly formed tissue) was calculated (Appendix A). All slides were independently scored by two investigators blinded to the experimental condition. The measurements of the two investigators were averaged for each section.

To investigate the presentation of neutrophils in the defect, immunohistochemistry for myeloperoxidase (MPO) was performed on retrieved bovine osteochondral samples from the mouse study. After dewaxing, antigen retrieval was performed by placing the slides with Tris/EDTA (pH9) in a water bath at 95 °C for 20 min. Then, the slides were pre-incubated with 10% normal rabbit serum (NRS, Invitrogen, Waltham, MA, USA) in PBS containing 1% bovine serum albumin (BSA, Sigma, Saint Louis, MI, USA) and 1% milk powder (ELK, Campina, Amersfoort, The Netherlands). The slides were incubated by the first antibody against MPO (Thermo Scientific, Waltham, MA, USA, 1:200 dilution) or rabbit IgG antibody (DakoCytomation, California, USA, 1:10,000 dilution) as the negative control in PBS containing 1% BSA for 1 h. Next, the slides were incubated by biotinylated goat α-rabbit (Biogenex, Fremont, CA, USA, 1:50 dilution in PBS containing 1% BSA and 5% mouse serum of total volume) for 30 min. Then, the reaction was amplified by streptavidin-labelled alkaline phosphatase (Biogenex, Fremont, CA, USA) diluted 1:50 in PBS containing 1% BSA and visualized by subsequent incubation of Neu Fuchsin substrate. Slides were counterstained with Hematoxylin.

To evaluate the infiltration of macrophages in the defect, immunohistochemistry for F4/80 was performed on the bovine osteochondral plugs retrieved from the in vivo mouse study. For antigen retrieval, each dewaxed slide was treated with 300 μL proteinase K (20 µg/mL, Thermo Scientific, Waltham, MA, USA) solution and incubated at 37 °C for 30 min. Then, the slides were pre-incubated with 10% NRS in PBS containing 1% BSA and 1% milk powder. The following steps were similar to the immunohistochemistry for F4/80, with the first antibody against F4/80 (eBioscience, San Diego, CA, USA, 1 μg/mL) or rat IgG2a (eBioscience, San Diego, CA, USA, 1 μg/mL) as the negative control in PBS containing 1% BSA; the second antibody: biotinylated rabbit anti-rat IgG (6 μg/mL in PBS containing 1% BSA and 5% mouse serum of total volume), and third antibody: streptavidin-labelled alkaline phosphatase (Biogenex, Fremont, CA, USA) diluted 1:50 in PBS containing 1% BSA. To distinguish between pro-inflammatory (M1) and anti-inflammatory/tissue-repair (M2) macrophages, immunohistochemistry for inducible Nitric Oxide Synthase (iNOS, as an indicator for pro-inflammatory M1 macrophages) was performed. The steps were similar to the immunohistochemistry for MPO, with the first antibody against iNOS (2 μg/mL, Abcam, Cambridge, UK).

The slides were ranked according to the positive degree of immunohistochemical staining, and all negatively stained sections were ranked 0. Only areas that were also stained for F4/80 were taken into account when the iNOS staining was ranked.

### 2.9. Statistical Analysis

All statistical tests were performed using SPSS software 28.0 (SPSS Inc., Chicago, IL, USA). The repair tissue volume was expressed as mean ± standard deviation (SD). The rankings of immunohistochemical MPO, F4/80, and iNOS staining were presented as column plots in graphs. Multiple comparisons between scaffold-only, BMP-2, and PDGF groups in bovine osteochondral plug samples were analysed by a Kruskal–Wallis test. Statistically significant differences between scaffold-only and scaffold + BMP-2 groups or between trochlear groove and femoral condyle groups in caprine samples were determined by a Mann–Whitney U test. A *p* value ≤ 0.05 was considered statistically significant.

## 3. Results

### 3.1. Scaffold Characterization

SEM images show the detailed morphology of the cartilaginous layer (collagen, Figure 1A) and the bone layer (60% collagen and 40% magnesium-hydroxyapatite, Figure 1B) of the scaffolds. The ratio between collagen and magnesium-hydroxyapatite is 69/31 *w/w*%. The mineral phase is composed of non-stoichiometric, calcium deficient, magnesium-substituted hydroxyapatite. The total porosity of the osteochondral scaffold was 83 ± 1%.

### 3.2. The Release Profiles of BMP-2 and PDGF-BB In Vitro

Of the added BMP-2, 0.6% was detected in the medium after adsorption for the collagen-only layer and 3.0% for the Col-Mg-HAp layer, indicating that most of the BMP-2 was indeed adsorbed and that both layers had a similar adsorption capacity at the tested volume. Over 14 days, only 48.8 ± 14.8 ng and 22.1 ± 3.4 ng of the adsorbed BMP-2 was released from the collagen-only layers or Col-Mg-HAp layers, respectively (Figure 2A). BMP-2 was largely retained within the scaffolds. A similar release pattern was observed for both layers, although the collagen-only layer released (2-fold) more over the 14-day period (Figure 2A).

Of the added PDGF-BB, 3.9% was detected in the medium after adsorption for the collagen-only layer and 5.9% for the Col-Mg-HAp layer, indicating that most of the PDGF-BB was adsorbed and that both layers had a similar adsorption capacity. In contrast to BMP-2, a rapid release of PDGF-BB from the collagen-only layer was observed, and 600.4 ± 273.6 ng was released within 6 h (Figure 2B). Interestingly, almost no release was observed from the Col-Mg-HAp layer; only 33.8 ± 11.8 ng PDGF-BB was cumulatively released from the Col-Mg-HAp layer over seven days.

### 3.3. Effect of BMP-2 and PDGF-BB on Cell Ingrowth in an Ex Vivo Culture Model

To evaluate the effect of BMP-2 or PDGF-BB addition onto the scaffold on cell recruitment from adjacent osteochondral tissues, an ex vivo model was used. After three weeks, the scaffolds filled the osteochondral defects, and cells infiltrated the scaffolds. In the scaffold without growth factors, cells were mostly located at the periphery of the scaffold (Figure 3A). Interestingly, when BMP-2 or PDGF-BB was added, cell infiltration was also observed in the centre, particularly in the collagen-only layer (Figure 3A). Almost no cells were found in the Col-Mg-HAp layer. Overall, the addition of growth factors significantly increased cell infiltration into the scaffolds (Figure 3B).

### 3.4. Effect of BMP-2 and PDGF-BB at the Early Phases of Bone Repair in an In Vivo Semi-Orthotopic Osteochondral Defect Model in Mice

One dose of BMP-2 (4 µg) and two doses of PDGF-BB (100 ng and 2 µg) were tested in order to investigate their potential effect on the early stages of bone repair. After four weeks, neither cartilage nor bone formation were observed in any of the samples with scaffold-only (Figure 4A). When BMP-2 or PDGF-BB was adsorbed into the scaffold, cartilage-like tissue was found in 1 out of 4 samples with 100 ng PDGF-BB, 2 out of 4 samples with 2 µg PDGF-BB and 2 out of 4 samples with 4 µg BMP-2 (Figure 4B). Interestingly, the scaffolds adsorbed with either 4 µg BMP-2 or 2 µg PDGF-BB showed less MPO (as an indicator for neutrophils) and iNOS (as an indicator for pro-inflammatory macrophages) staining; albeit, this did not reach statistical significance due to the relatively low sample size (Appendix A).

The effect of 4 µg BMP-2, 2 µg PDGF-BB, or the combination of the two were further investigated in the semi-orthotopic mouse model after eight weeks. Inflammation was largely resolved in all samples. Immunohistochemical staining for MPO and iNOS was negative in all but one defect that was from the scaffold-only group. No difference was found between the different groups. All defects were filled with bone tissue, and blood vessels were observed, independent of the presence of growth factors (Figure 4C). Overall, there was slightly more tissue repair in BMP-2-adsorbed scaffolds (78.9 ± 23.1% of the defect filled) compared with PDGF-BB-adsorbed scaffolds (50.9 ± 28.0%) or scaffold-only (68.8 ± 36.2%) (Figure 4D). Partially repaired bone defects were found in 5 out of 7 defects fitted with PDGF-BB-adsorbed scaffolds, while only in 3 out of 7 in the scaffold-only group and 2 out of 7 in the group of BMP-2-adsorbed scaffolds.

### 3.5. Effect of BMP-2 on Bone Repair in an In Vivo Caprine Model

#### 3.5.1. Scaffold Implantation and Clinical Observations

Most of the osteochondral tissue was regenerated in the osteochondral defects when 4 µg BMP-2 was adsorbed to the Col/Col-Mg-HAp scaffold in the mouse model. Therefore, BMP-2 was selected to be further investigated in an established bilateral preclinical caprine osteochondral defect model. Two differently loaded locations were selected in this caprine model: a weight-bearing region of the medial femoral condyle and a non-weight-bearing area in the trochlear groove. During surgery, scaffolds were successfully press fit into osteochondral defects flush with the surrounding cartilage. All defects bled after drilling, and scaffolds became saturated with fresh blood upon implantation, as observed by a change in colour of the implanted scaffold.

Recovery was uneventful and no postoperative complications occurred except for the unforeseen death of one animal three days post-surgery due to ruminal acidosis caused by overeating of concentrates, unrelated to the experimental procedure. Macroscopic appearance after three days in the unforeseen dead animal showed that scaffolds were stable in the osteochondral defects (Appendix A). The two layers of the scaffold were clearly visible in the defects (Appendix A).

Clinical examination of animals daily for 14 days post-surgery and weekly until the endpoint at six months demonstrated excellent recovery from surgery and a normal pain-free range of movement and normal locomotion from 3–10 days post-surgery. After six months, there were no signs of inflammation or cartilage abnormalities found during a post-mortem evaluation of the joints. No signs of joint swelling, effusion, abnormal mobility, synovial adhesions, synovial fluid and membrane abnormalities, abnormal wound healing, patellar luxation, or erosions or lesions on the opposite cartilage surface were found in any of the animals.

#### 3.5.2. Tissue Repair in the Non-Weight-Bearing Trochlear Groove Osteochondral Defects

To quantify the subchondral bone formation within the bone defect, micro-CT analysis was performed. Well-repaired subchondral bone was observed in the images of the trochlear groove treated with both the scaffold-only and BMP-2-adsorbed scaffold at six months (Figure 5A). A slightly higher Tb.N was found in the defects with BMP-2-adsorbed scaffolds. No significant differences in the BV/TV, Tb.Th, Tb.N, or Tb.Sp were found between the groups (Figure 5B).

Macroscopical and histological evaluation of the cross sections of the defect supported the micro-CT quantification (Figure 5C). Overall, the scaffolds were completely degraded, and a well-structured subchondral trabecular bone was observed in most bone defects after six months (Figure 5C). Slightly more bone tissue (93.9 ± 4.4% vs. 90.1 ± 8.4%) and less fibrous tissue (4.0 ± 4.5% vs. 7.8 ± 8.4%) was found in the bone defects when the scaffolds were adsorbed with BMP-2 compared with the scaffold-only, although no significant difference was reached (Figure 5D). Notably, a structure without trabecular bone but with only bone marrow was observed underneath some defects, independent of the condition.

Macroscopically, defects were covered with newly formed cartilage with good integration into the surrounding native tissue. Small, scattered fissures or cracks were observed on the surfaces of some defects, and no noticeable depressions were observed (Appendix A). The ICRS and Goebel scores for the scaffold-only group had a median score of 11.3 ± 0.5 and 19.0 ± 0.7, respectively (Appendix A). For the scaffold + BMP-2 group, the macroscopic ICRS and Goebel scores were 11.6 ± 0.6 and 19.3 ± 0.5, respectively (Appendix A). All the trochlear groove samples were classified as normal (grade I) or nearly normal (grade II) cartilage. Overall, no significant difference was observed in cartilage repair between scaffold-only and BMP-2-adsorbed scaffold. On histology, 43.2 ± 22.1% (scaffold-only) and 47.4 ± 19.8% (scaffold + BMP-2) of the newly formed tissue generated in the cartilage defects was fibrous tissue after six months as indicated by predominantly Picrosirius Red instead of Alcian Blue staining in RGB staining (Figure 4E). Moreover, only 32.4 ± 27.6% (scaffold-only) and 35.2 ± 20.6% (scaffold + BMP-2) of the repaired tissue in the cartilage region was Alcian Blue-positive, indicating hyaline cartilage (Figure 5E).

#### 3.5.3. Tissue Repair in the Weight-Bearing Femoral Condyle Osteochondral Defects

For the femoral condyle, micro-CT analysis was performed in the same manner as for the trochlear groove. In the weight-bearing femoral condyle, well-structured subchondral bone was formed after six months in defects treated with the scaffold-only or the BMP-2-adsorbed scaffold (Figure 6A), no significant difference was found in the BV/TV and Tb.Th parameters (Figure 6B). Surprisingly, a slightly lower BV/TV was found in defects with BMP-2-adsorbed scaffolds, indicating relatively worse bone repair when BMP-2 was adsorbed. A significantly smaller Tb.N (1.1 ± 0.4 vs. 1.5 ± 0.2, *p* = 0.019, Figure 6B) and a greater Tb.Sp (1.2 ± 0.6 vs. 0.7 ± 0.1, *p* = 0.015, Figure 5B) were observed in the femoral condyle bone defects fitted with BMP-2-adsorbed scaffolds, suggesting a relatively courser bone structure when BMP-2 was adsorbed.

Macroscopic and histological evaluation of the cross sections through the defect supported the micro-CT quantification (Figure 6C). Newly formed hyaline cartilage-like tissues were mostly supported by a well-structured subchondral trabecular bone that was well integrated in the surrounding native bone (Figure 6C). Surprisingly, significantly less cartilage and bone tissue were observed in the bone defects fitted with BMP-2-adsorbed scaffolds compared with the bone defects treated with scaffold-only (92.7 ± 9.5% vs. 99.1% ± 1.7%, *p* = 0.043, Figure 6D). In addition, significantly more fibrous tissue was found in the bone defects when BMP-2 was adsorbed onto the Col/Col-Mg-HAp scaffolds (6.9 ± 8.4% vs. 1.0 ± 1.6%, *p* = 0.035, Figure 6D). Some defects demonstrated cyst-like areas without trabecular bone below some of the defects. However, marrow-like tissue was found in these areas.

The defects were mostly covered with hyaline cartilage-like tissue with macroscopically good integration with the surrounding native tissue (Appendix A). Small, scattered fissures or cracks were observed on surfaces of some defects, and no noticeable depressions were observed except for one sample treated with Col/Col-Mg-HAp scaffold-only. The median ICRS score was 10.8 ± 0.5 out of 12, and the median Goebel score was 18.9 ± 0.5 out of 20 (Appendix A) when scaffold-only was placed in the defects. The defects with the scaffold + BMP-2 received median ICRS scores of 10.3 ± 1.8 out of 12 and median Goebel scores of 18.5 ± 1.8 out of 20 (Appendix A). All the samples were classified as normal (grade I) or nearly normal (grade II) cartilage except for one sample treated with BMP-2 (grade III). By histological examination, the scaffolds were completely degraded after six months, and round cells residing within lacunae were present in the cartilage region. GAG and collagen were present in the repair tissue in the defects demonstrated by RGB staining (Figure 6C), indicating cartilaginous tissue formation. In fact, 77.8 ± 16.5% (scaffold-only) and 78.8 ± 14.7% (scaffold + BMP-2) of the newly formed tissue was Alcian Blue-positive, indicating hyaline-like cartilage (Figure 6E).

#### 3.5.4. Different Tissue Repair in the Non-Weight-Bearing Trochlear Groove and the Weight-Bearing Femoral Condyle Osteochondral Defects

In trochlear groove bone defects, the trabecular number was smaller (1.2 ± 0.5 vs. 1.5 ± 0.2, *p* = 0.063) and the trabecular separation was greater (1.1 ± 0.6 vs. 0.7 ± 0.1, *p* = 0.052) compared with the femoral condyle defects when scaffold-only was implanted, although no significant difference was found. Interestingly, when BMP-2 was adsorbed onto the scaffold, an opposite trend was found in the Tb.N (1.4 ± 0.4 in trochlear groove and 1.1 ± 0.4 in femoral condyle, *p* = 0.143) and the Tb.Sp (0.8 ± 0.5 in trochlear groove and 1.2 ± 0.6 in femoral condyle *p* = 0.105). In other words, there was a 0.2 ± 0.6 greater Tb.N and a 0.3 ± 0.9 smaller Tb.Sp in the trochlear groove defect versus a 0.47 ± 0.6 smaller Tb.N (*p* = 0.029, compared with trochlear groove) and a 0.5 ± 0.7 greater Tb.Sp (*p* = 0.063, compared with trochlear groove) in the femoral condyle defect when BMP-2 was adsorbed onto the scaffold, indicating that bone repair was improved in trochlear groove defects and reduced in femoral condyle defects when BMP-2 was added. Histological results further confirmed that significantly more bone-like tissue was regenerated in the trochlear groove defects compared with the femoral condyle defects when BMP-2 was added (93.9 ± 4.4% vs. 83.5 ± 9.8%, *p* = 0.011), while no difference between weight-bearing and non-weight-bearing sites was observed when scaffold-only was implanted (90.1 ± 8.4% vs. 86.7 ± 11.5%, *p* = 0.579). For cartilage repair, however, significantly less hyaline cartilage-like tissue was observed in the non-weight-bearing trochlear groove compared with the weight-bearing femoral condyle defects with either scaffold-only (32.4 ± 27.6% vs. 77.8 ± 16.5%, *p* = 0.003) or BMP-2-adsorbed scaffold (35.2 ± 20.6% vs. 78.8 ± 14.7%, *p* < 0.001).

## 4. Discussion

In this study, we evaluated the effectiveness of growth factor adsorption onto a bi-layered Col/Col-Mg-HAp scaffold in osteochondral defect repair. In vitro release results showed that the Col-Mg-HAp (bone) layer retained more growth factor than the collagen-only (cartilage) layer and that BMP-2 was retained much better than PDGF-BB. In an ex vivo osteochondral defect model, cell ingrowth into the scaffold was enhanced by BMP-2 and by PDFG-BB. In a semi-orthotopic non-weight-bearing osteochondral defect mouse model representing the early phase (four weeks) of defect repair in vivo, the addition of growth factors resulted in fewer pro-inflammatory cells and better tissue repair, with BMP-2 showing the most favourable results. Therefore, BMP-2 addition was taken forward for testing in an established preclinical large animal osteochondral defect model to study scaffold enhancement by BMP-2 in a physiological environment with different loading conditions. After six months in goats, both the scaffold-only and the BMP-2-adsorbed scaffold induced good osteochondral defect healing. Surprisingly, the addition of BMP-2 led to worse bone repair in the weight-bearing femoral condyle osteochondral defects, whereas this negative effect of BMP-2 was not seen in the non-weight-bearing trochlear groove osteochondral defect location.

Our approach was based on the sequential use of osteochondral defect models: the first experiments were performed in an ex vivo bovine osteochondral explant model, followed by a semi-orthotopic non-weight-bearing model where we implanted bovine osteochondral explants subcutaneously in mice. This took into account the effects of the innate immune system and blood vessel invasion. We then selected the most promising condition to be further evaluated in a preclinical caprine osteochondral defect model, where we tested both weight-bearing and non-weight-bearing repair conditions simultaneously. All models were developed and used previously to evaluate new osteochondral repair approaches [34,40]. In vivo animal models can closely resemble the human osteochondral microenvironment in the context of the presence of immune cells and tissue repair factors. The mouse model allows the screening of four conditions in one animal, whereas the caprine bilateral osteochondral defect model is a fully immune-competent model using outbred animals. That model also allows within-animal controls (comparing left and right knees), and provides the opportunity to assess tissue regeneration in both weight-bearing and non-weight-bearing locations within the same joint. These animal models offered the opportunity to investigate the possible effect of incorporating growth factors into a Col/Col-Mg-HAp scaffold for osteochondral repair towards translation into the human patient.

Loading can have a significant effect on tissue repair [41,42]. Not much is known about the interaction between mechanical loading and growth factors and their effect upon osteochondral tissue repair [43,44]. To study the effect of mechanical stimuli on osteochondral repair during normal ambulation, we used both a non-weight-bearing location (a distal region in the trochlear groove) and a weight-bearing location (a central region on the medial femoral condyle) in the bilateral stifle caprine osteochondral defect model [45,46]. Better subchondral bone repair was observed in the non-weight-bearing trochlear groove defects than in the weight-bearing femoral condyle defects when BMP-2 was adsorbed onto the scaffold, while no significant difference in repair was found between locations when the scaffold without additions was evaluated. In the non-weight-bearing trochlear groove defects, there was a trend towards more bone-like tissue being generated in the BMP-2-adsorbed scaffolds compared with the scaffold-only. However, this difference did not reach statistical significance, which might be due to the excellent repair capacity of the scaffold-only. The weight-bearing femoral condyle defects implanted with BMP-2-adsorbed scaffolds appeared to have less bone repair and more fibrous tissues in the bone defect compared with the scaffold-only.

Notably, there was no mechanical loading in the semi-orthotopic mouse model and at the four-week time point, the addition of BMP-2 to the scaffolds seemed beneficial. Our results reveal the interesting hypothesis that, with a specific dose of BMP-2, the beneficial effect of BMP-2 on bone repair is apparent in non-weight-bearing conditions, whilst BMP-2 addition can be detrimental to bone repair in weight-bearing conditions. Although the effect of BMP-2 on osteochondral defect repair in different loading environments is still unknown, the action of BMP-2 on tibia fractures or femoral bone defect healing was demonstrated to be dependent on the mechanical environment in vivo [47,48]. This might be due to an interaction between BMP-2 and mechanical loading [49] and related to the dosage used. Compression and loading affect BMP signalling both immediately and in a long-term manner [50]. Mechanical loading was shown to increase BMP-2 expression [51], and the effect of BMP-2 can be strongly potentiated by mechanical forces [49]. It is possible that the combination of added and locally produced BMP-2 might lead to an overstimulation of BMP-2 signalling, which is shown to cause inflammation, bone resorption, and fibrotic tissue formation [52,53,54]. No previous study has reported this potential side effect nor the relationship with mechanical loading. However, the mechanical loading patterns in long bone defects with fixation and osteochondral defects with scaffolds in the femoral condyle (especially with the presence of synovial fluid) are quite distinct from one another. Therefore, further studies are needed to elucidate if and at which step of the BMP signalling cascade the pathway is modulated and by which type of mechanical stimuli.

BMP-2 is, to date, the only FDA-approved and commercially available osteoinductive growth factor used in clinics. The function and application of BMP-2 in promoting bone regeneration and bone remodelling has been widely investigated preclinically and clinically [55,56,57]. Mg^2+^ was incorporated in this scaffold, which might upregulate the bioactivity of BMP-2 upon calcium phosphate cement via enhanced BMP receptor recognition [58]. Previous studies have demonstrated that BMP-2 accelerated the migration of bone marrow mesenchymal stromal cells (MSCs) in vitro and in vivo [59,60]. In the presented study, we also observe the promotion of cell ingrowth by BMP-2 addition in our ex vivo culture model. Although the primary expected functions of BMP-2 in promoting bone repair are to enhance MSC migration to the sites and differentiation into osteoblasts and to enhance the osteogenic capacity of osteoblasts, BMP-2-induced osteogenesis may also involve an immunoregulatory role [61,62]. Macrophages act as immune cells and osteoclast precursors and are involved in multiple stages of bone healing [63]. BMP-2 might diminish the expression of pro-inflammatory phenotypic markers and promote the macrophage transition towards a more tissue repair-like phenotype [62]. Neutrophils and M1 macrophages participate in tissue repair as effector cells in inducing inflammation, and M2 macrophages are involved in the resolution of inflammation, promoting angiogenesis, and matrix remodelling [64]. In our semi-orthotopic model, fewer M1 macrophages, as well as neutrophils, were found in the BMP-2 condition than in the scaffold-only condition after four weeks. Therefore, the regulatory effect of BMP-2 in a local osteoimmune environment might be one of the potential mechanisms for promoting bone healing in the early phase.

Although BMP-2 has been clinically applied because of its osteogenic effect, it is still not widely used due to the adverse events associated with implanted supraphysiological high doses [57]. The most documented side effect is ectopic ossification. Aulin et al. demonstrated that intra-articular injection of a hyaluronan hydrogel containing BMP-2 (150 μg/mL) resulted in excessive ectopic bone formation on the knee joint surface of rabbits (6–7-month-old females) [65]. In our study, we used 57.1 μg/mL BMP-2 for both 4 mm × 4 mm and 6 mm × 6 mm cylindrical osteochondral defects according to our previous in vivo results [30]. HAp was reported to have a high affinity for BMP-2 due to the large surface area and functional groups [29,66]. Three types of functional groups, −OH, −NH_2_, and −COO, allow BMP-2 to adsorb on the HAp surface [66]. This is not the case for all growth factors, where, for example, no functional groups or only one might be present. In vitro, we demonstrated that BMP-2 was bound and largely retained in both layers, especially in the Col-Mg-HAp layer. In this way, a sustained release was expected, thereby reducing the adverse effects of BMP-2. Over longer periods, however, correlations between the in vitro and in vivo settings become ever more unreliable since the release in vivo will be determined by the degradation of the scaffold as well. In fact, no BMP-2-related ectopic ossification or abnormal inflammation was observed in the knee joints of our experimental goats macroscopically or on micro-CT. Similarly, no ectopic ossification was reported in a publication that investigated the addition of 625 μg/mL of BMP-2 (adsorbed onto Col/HAp scaffolds) in skeletally mature male rabbits [29].

Consistent with other studies, we found that PDGF-BB is an effective chemoattractant of cells ex vivo [31,67]. PDGF-BB was rapidly released from the collagen-only layers within six hours; most of the PDGF-BB was retained in the Col-Mg-HAp layers in vitro. This might be due to the capacity of HAp in adsorbing a large quantity of proteins and drugs. At the early repair phase in the in vivo mouse model, both dosages (1 μg and 100 ng) of PDGF-BB slightly improved tissue formation. Most of the regenerated tissue was in the bone defect area, which might be related to the sustained release from the Col-Mg-HAp layers, which is aimed at the repair of the bone defect specifically. Lee et al. reported that labelled cells migrated towards the osteochondral defect when defects were treated with PDGF-AA or PDGF-BB-loaded heparin-conjugated fibrin [67]. Overall, the chemotactic ability of PDGF-BB might be one of the mechanisms in inducing tissue repair in the early phase. However, after eight weeks, the defects loaded with PDGF-BB generated slightly less osteochondral tissue compared with scaffold-only. A previous study on osteochondral repair also showed that the addition of 1 μg/mL PDGF-BB worsened the cartilage repair in an in vivo subcutaneous mouse model, although cell recruitment was enhanced in vitro. The short half-life of PDGF might be one of the reasons for the different results between in vitro and in vivo studies. Zhang et al. demonstrated that PDGF-BB overexpression improved the osteogenic and angiogenic abilities of MSCs in a critical-sized rat calvaria defect model [68].

We also expected the combination of BMP-2 and PDGF-BB to further improve bone repair, since PDGF was reported to modulate BMP-2-induced osteogenesis in periosteal progenitor cells [69]. However, no significant improvement was found after eight weeks in the semi-orthotopic mouse model in vivo. Therefore, PDGF-BB and the combination of BMP-2 and PDGF-BB were not further evaluated in our large animal osteochondral defect model.

Suboptimal subchondral bone repair of osteochondral defects might lead to damage of the renewed overlying cartilage in the long term [16,17,18]. In our caprine large animal model, a blinded macroscopic evaluation of repaired cartilage tissue indicated the presence of a smooth white cartilaginous layer that was continuous with surrounding naive cartilage, both with control scaffolds and BMP-2-enhanced scaffolds. Histology further confirmed the presence of hyaline cartilage in the superficial layer, with no significant improvement or deterioration observed when BMP-2 was added. This might be due to the fact that tissue repair was assessed at only one, relatively early, six-month time point in the caprine model. A longer 12- or 24-month study might be useful to confirm the potential effectiveness of enhanced bone repair on the long-term survival of the neo-cartilage.

## 5. Conclusions

Adsorption of BMP-2 onto a Col/Col-Mg-HAp scaffold reduced bone formation in weight-bearing osteochondral defects but not in non-weight-bearing osteochondral defects. Since the application of BMP-2 adsorbed to a Col/Col-Mg-HAp scaffold in osteochondral defects did not lead to adverse effects in the joint of goats, and human patients are not allowed full weight-bearing in the first weeks after the osteochondral repair, further investigation is warranted, taking into consideration the dose of BMP-2, timing, and location of application for osteochondral defect repair.

## Figures and Tables

**Figure 1 jfb-14-00111-f001:**
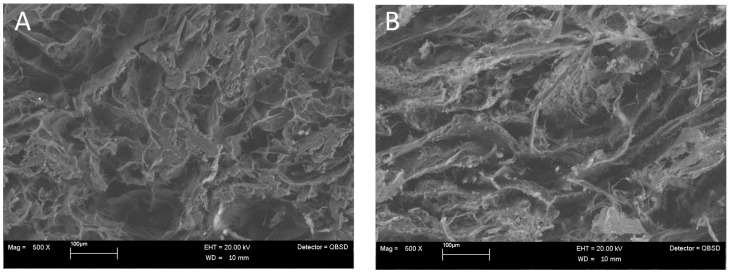
SEM images of (**A**) the cartilaginous layer (collagen) and (**B**) the bone layer (60% collagen and 40% magnesium-hydroxyapatite).

**Figure 2 jfb-14-00111-f002:**
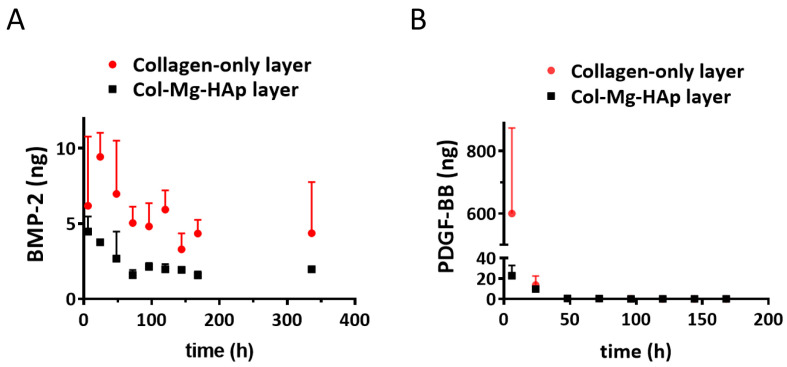
Bone morphogenetic protein 2 (BMP-2) and platelet-derived growth factor-BB (PDGF-BB) released from the different layers of the Col/Col-Mg-HAp scaffold. (**A**) In vitro release of BMP-2 from the different layers of Col/Col-Mg-HAp scaffold over a 14-day period. (**B**) In vitro release of PDGF-BB from the different layers of the Col/Col-Mg-HAp scaffold over a 7-day period. The release of BMP-2 and PDGF-BB is presented as the released dose at each time point. Data points indicate the mean ± SD of 3 samples per time point.

**Figure 3 jfb-14-00111-f003:**
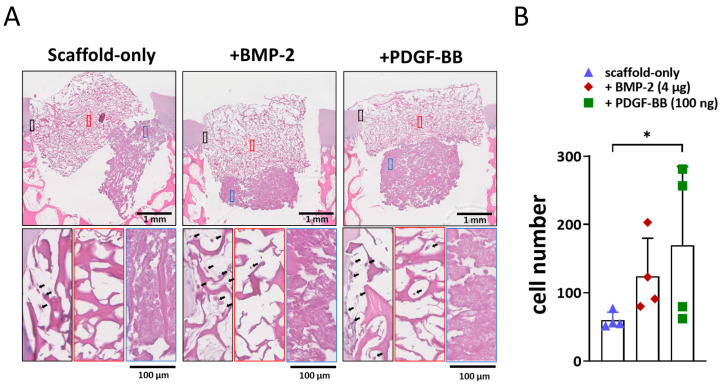
Adsorption of BMP-2 or PDGF-BB might improve the cell recruitment property ex vivo. (**A**) Representative images of the 3-week constructs stained with H&E. Scale bars indicate 1 mm and 100 µm, respectively. Magnified images showed cell infiltration at the periphery (black square), in the centre (red square) of the collagen-only layer, and in the Col-Mg-HAp layer (blue square) of the scaffold. Black arrows indicate infiltrated cells. (**B**) The number of cells infiltrated into the scaffolds. Each bar indicates the mean ± SD of 4 samples per condition. * *p* < 0.05 analysed by a Kruskal–Wallis test.

**Figure 4 jfb-14-00111-f004:**
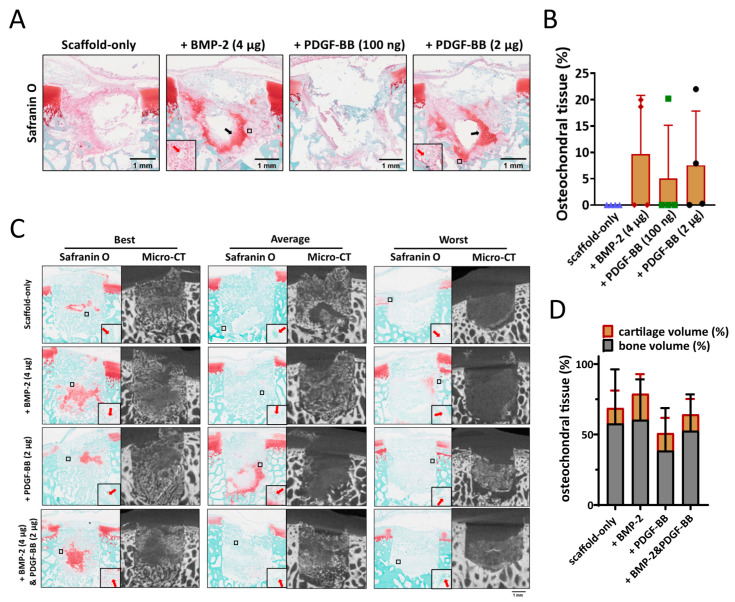
The effect of BMP-2 or PDGF-BB adsorption onto Col/Col-Mg-HAp scaffolds on osteochondral defect repair in a semi-orthotopic model in vivo. (**A**) Representative images of the 4-week repair constructs stained with Safranin O. Scale bars indicate 1 mm. Black arrows indicate cartilage-like tissue. Red arrows indicate blood vessels. (**B**) The percentage of the defect filled with osteochondral tissue (%) in the osteochondral defects at 4 weeks. (**C**) Representative images of the 8-week repair constructs stained with Safranin O, and the 8-week CT images. The best, average, and worst repaired samples are presented based on osteochondral tissue volume (%). Scale bars indicate 1 mm. Red arrows indicate blood vessels. (**D**) The percentage of the defect filled with osteochondral tissue (%) in the osteochondral defects at 8 weeks. Scale bars indicate 1 mm.

**Figure 5 jfb-14-00111-f005:**
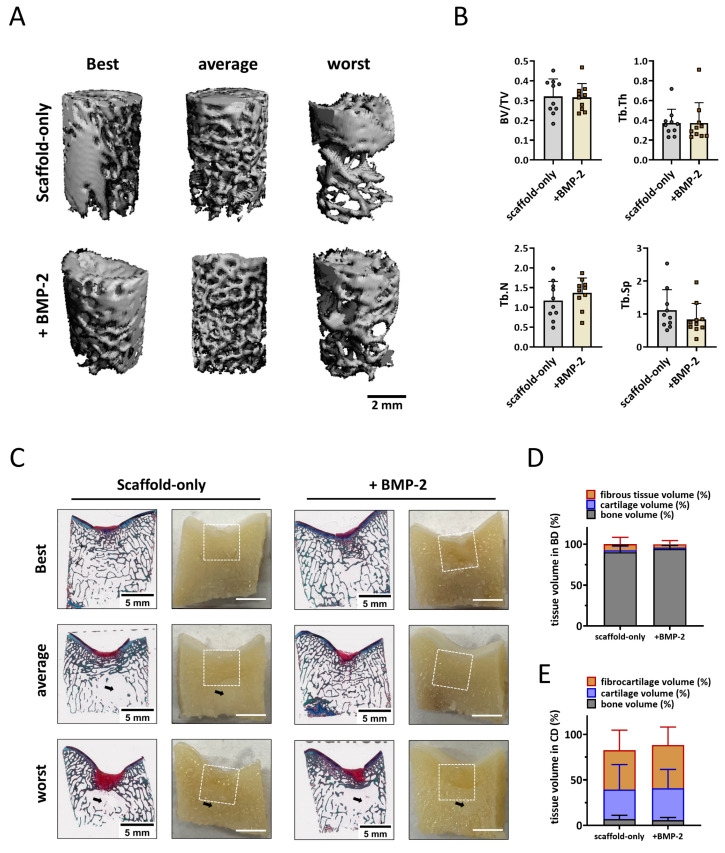
Tissue repair in non-weight-bearing trochlear groove defects. (**A**) Representative micro-CT reconstructions treated with either scaffold-only or scaffold adsorbed with BMP-2. The best, average, and worst repaired samples are presented based on BV/TV. The scale bar indicates 2 mm. (**B**) BV/TV, trabecular thickness (Tb.Th [mm]), trabecular number (Tb.N [1/mm]), and trabecular separation (Tb.Sp [mm]) in the bone defects after 6 months. (**C**) RGB (Alcian Blue, Fast Green, and Picrosirius Red) staining and macroscopic cross-sectional view of osteochondral defects treated with either scaffold-only or scaffold adsorbed with BMP-2. The best, average, and worst repaired samples are presented. White squares indicate 6 ∗ 6 mm osteochondral defects. Black arrows indicate the structure with only bone marrow. The scale bar indicates 5 mm. (**D**) The percentage of tissue volume calculated in the bone defects (BD). (**E**) The percentage of tissue volume calculated in the cartilage defects (CD).

**Figure 6 jfb-14-00111-f006:**
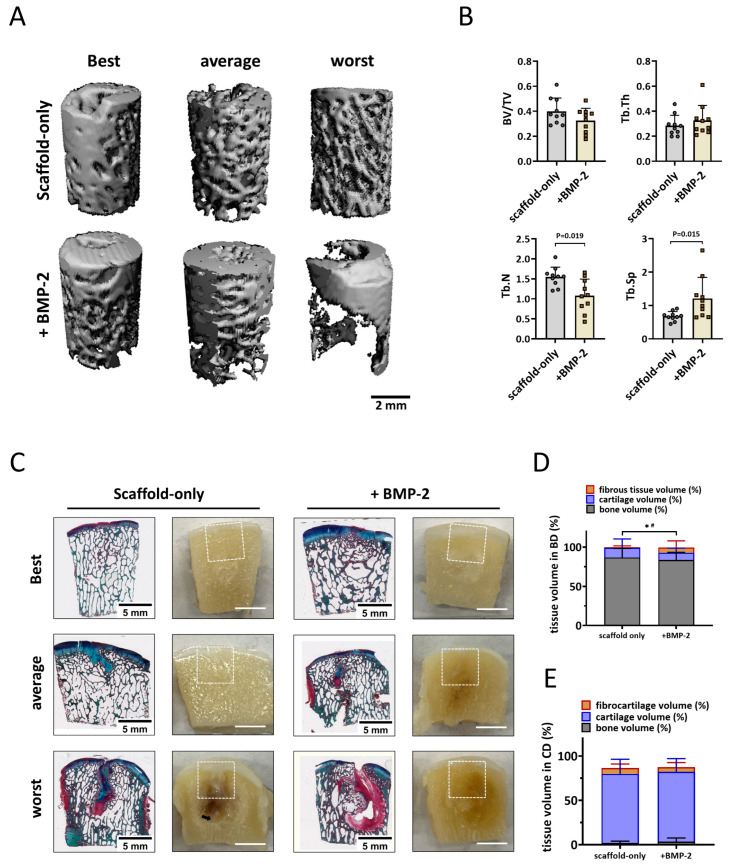
Bone repair in the weight-bearing femoral condyle defects deteriorated with the adsorption of BMP-2 onto the scaffold. (**A**) Representative micro-CT reconstructions treated with either scaffold-only or scaffold adsorbed with BMP-2. The best, average, and worst repaired samples are presented based on BV/TV. The scale bar indicates 2 mm. (**B**) BV/TV, trabecular thickness (Tb.Th [mm]), trabecular number (Tb.N [1/mm]), and trabecular separation (Tb.Sp [mm]) in the bone defects after 6 months. Tissue repair in the femoral condyle defects. (**C**) RGB (Alcian Blue, Fast Green, and Picrosirius Red) staining and macroscopic images of osteochondral defects treated with either scaffold-only or scaffold adsorbed with BMP-2. The best, average, and worst repaired samples are presented. White squares indicate 6 ∗ 6 mm osteochondral defects. The scale bar indicates 5 mm. (**D**) The percentage of tissue volume calculated in the bone defects (BD). * *p* < 0.05 in fibrous tissue, # *p* < 0.05 in osteochondral (cartilage-like and bone-like) tissue. (**E**) The percentage of tissue volume calculated in the cartilage defects (CD).

## Data Availability

Data is contained within the article or Appendix A.

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
