# Peer review of "Effectiveness of BMP-2 and PDGF-BB Adsorption onto a Collagen/Collagen-Magnesium-Hydroxyapatite Scaffold in Weight-Bearing and Non-Weight-Bearing Osteochondral Defect Bone Repair: In Vitro, Ex Vivo and In Vivo Evaluation"

_jfb, 2023, doi:10.3390/jfb14020111_

Round 1

Reviewer 1 Report

In the article entitled “Effectiveness of BMP-2 and PDGF-BB adsorption onto a collagen/collagen-magnesium-hydroxyapatite scaffold in weightbearing and non-weightbearing osteochondral defect bone repair: in vitro, ex vivo and in vivo evaluation” the authors have shown the positive and negative effect of these growth factors using various models.

There are several studies in which  BMP-2/PDGF  have been evaluated in bone therapy. Apart from the dose, the timing of its delivery, the type of carrier used for delivery etc., play a crucial role in the outcome. Therefore the rationale behind the use of a particular dose, the time of implantation as well as the delivery system should be explained with respect to the literature. The authors have described the results but have failed to provide any explanation. The results have to be compared with the reported literature with suitable justification.

AS the outcome on bone regeneration is dose dependent, the long term release profile of biologically active growth factors has to be established.  A one week study is insufficient.

SEM images should be provided of the scaffold.

The authors should explain the reason for better adsorption of growth factors in collagen vis-a-vis collagen/Mg-hydroxyapatite. If the scaffold is bilayered, how did the authors differentiate the adsorption capacity?

Some of the statements are very confusing. For example:

In section 3.2 After 3 weeks, the scaffolds were clearly filling the osteochondral defects and cells infiltrated the scaffolds, especially in the collagen-only layers (Figure 2). In the scaffold without growth factors, cells were mostly located at the periphery of the scaffold (Figure 2A). Interestingly, when BMP-2 or PDGF-BB was added, cell infiltration was also observed in the center of the collagen layer (Figure 2A), although almost no cells were found in the Col-Mg-HAp layer. Overall, the addition of growth factors significantly increased cell infiltration into the scaffolds (Figure 2B).

In Fig 2 A, the highlighted area shows the peripheral area of the scaffold. How do the authors claim infiltration in the center of the scaffold??

Data supporting the claim that no cell infiltration was observed in Col-Mg-HAp layer is not provided.

In comparison to the ex-vivo model in which no cell infiltration was seen in the Col-Mg-Hap, in vivo semi- orthotopic osteochondral defect model exhibited cartilage and bone formation, suggesting cell migration. Why such a difference??

What is the fate of the scaffold following implantation? When does the scaffold degrade?

It is good that the authors have compared the osteochondral response in a goat model following the implantation of the respective scaffolds in a load-bearing and non-load bearing area.

The claim that the non-weightbearing trochlear groove defects implanted with BMP-2-adsorbed scaffolds generated more bone-like tissue compared to the scaffold-only, although no significant difference was found which might be due to the excellent repair capacity of the scaffolds -The statement is confusing.

The authors claim that the scaffold implanted with BMP-2 in weightbearing femoral condyle defects - have less bone repair and more fibrous tissues. How was this fibrous tissue evaluated??

Vascularization plays an important role in bone regeneration. However no study was done to evaluate the extent of vascularization. The sections have to be tested for type H vessels.

It is recommended that the manuscript should be revised for better clarity.

Reviewer 2 Report

I have evaluated the manuscript “Effectiveness of BMP-2 and PDGF-BB adsorption onto a collagen/collagen-magnesium-hydroxyapatite scaffold in  weight-bearing and non-weight bearing osteochondral defect bone repair: in vitro, ex vivo and in vivo evaluation.” The manuscript is very interesting to read and the results were clearly presented. Moreover, this study can be a potential approach for improving bone repair in non-weight-bearing defects. However, the authors should address a few points before publication in the journal of Functional Biomaterials.

The novelty of the manuscript is not apparent. What is the role of magnesium in BMP-2 and PDGF-BB adsorption is not clear.

The scientific importance of using a magnesium-hydroxyapatite scaffold is missing in the introduction part.

Details about the collagen/collagen-magnesium-hydroxyapatite scaffold are not included. How is it prepared? 

Details about the crystalline phase, purity, pore characteristics, elemental composition, and the ratio between collagen and hydroxyapatite as well as magnesium, are not included. Because crystallographic properties, trace elements, as well as nanoscale properties, plays a vital role in bone absorption and desorption.

“Weightbearing” may be written as weight bearing or weight bearing. Clarify.

The adverse effects of adding BMP-2 in the weight-bearing situation are not apparent.  Provide scientific reason.

PDGF-BB was burst released from the collagen-only layer, whereas BMP-2 was retained mainly in both layers. Why? Because HAp is a ceramic matrix, and the burst release of growth factors was observed from HAp by many researchers. Clarify.

Reviewer 3 Report

Thank you for your submission. Actually, subchondral bone repair still is a challenge in the clinic. A vast array of biomaterials have been developed for tissue regeneration. Particularly, scaffolds are widely used and exhibit excellent properties in biomedicine. Furthermore, bioactive factors are introduced into the scaffolds to promote tissue regeneration. This article evaluated the osteogenesis of BMP-2 or PDGF-BB loaded Col/Col-Mg-HAp scaffold. Moreover, the scaffolds have been implanted into the non-weightbearing and weightbearing areas for further study. However, several limitations need to be revised before publication:

1.     In the introduction and discussion, BMP-2 and PDGF-BB were beneficial for bone osteogenesis, however, in the results, the BMP-2 group have not promoted bone regeneration, and then, in the discussion, the reasons have not been listed.

2.     In section 3.1, it has mentioned that “Only 0.6% of BMP-2 was not adsorbed after adsorption for the collagen-only layer and only 3.0% for the Col-Mg-HAp layer”, but the efficiency of BMP-2 absorption has been measured.

3.     In fig.3, there was little osteochondral tissue in the osteochondral defects at 4 weeks in the scaffold-only group, while the osteochondral tissue has been regenerated at 8 weeks. It seems that the rate of tissue repair is better in scaffold-only group, compared with other groups. The BMP-2 & PDGF-BB group has been examined at a late stage, however, the results of the early stage have not been shown. Moreover, there was a considerable difference within the group, the results may be not correct. For example, there are only 2 out of 4 samples with 4 µg BMP-2.

4.     In fig.4, there is no significant difference was observed in the non-weightbearing bone defects and defects between scaffold-only and BMP-2-adsorbed scaffold. However, the conclusion is “Adsorption of BMP-2 onto a Col/Col-Mg-HAp scaffold demonstrated promise to enhance bone repair in non-weightbearing osteochondral defects”.

5.     Some spelling mistakes should be corrected, for example, “CO2” in line 115.

6.     Most recent studies about tissue repair, such as Adv Fun Mater. 2021, 2009432; Bone Res. 2021, 33731688; Adv Fun Mater. 2022, 2202470; Biomater Transl. 2020, 35837659; Biomater Transl. 2022, 36105564; etc., are recommended to be cited in proper places.

Round 2

Reviewer 1 Report

None

Reviewer 3 Report

Thank you for your submission. Recently, numerous biomaterials have been developed for bone tissue engineering. The concerns that we proposed are all properly addressed. Now this revised manuscript could be accepted for publication.